# The antigenic landscape of human influenza N2 neuraminidases from 2009 until 2017

João Paulo Portela Catani[1,2]*, Anouk Smet[1,2], Tine Ysenbaert[1,2], Marnik Vuylsteke[3], Guy Bottu[4], Janick Mathys[4], Alexander Botzki[4], Guadalupe Cortes-Garcia[5], Tod Strugnell[5], Raul Gomila[5], John Hamberger[5], John Catalan[5], Irina V Ustyugova[5], Timothy Farrell[5], Svetlana Stegalkina[5], Satyajit Ray[5], Lauren LaRue[5], Xavier Saelens[1,2]*, Thorsten U Vogel[5]

[1]VIB-UGent Center for Medical Biotechnology, Ghent, Belgium; [2]Department of Biochemistry and Microbiology, Ghent University, Ghent, Belgium; [3]Gnomixx, Melle, Belgium; [4]VIB Bioinformatics Core, Ghent, Belgium; [5]Sanofi, Research North America, Cambridge, United States

*For correspondence:
joao.portelacatani@vib-ugent.
be (JPPC);
Xavier.Saelens@vib-Ugent.be
(XS)

**Abstract** Human H3N2 influenza viruses are subject to rapid antigenic evolution which translates into frequent updates of the composition of seasonal influenza vaccines. Despite these updates, the effectiveness of influenza vaccines against H3N2-associated disease is suboptimal. Seasonal influenza vaccines primarily induce hemagglutinin-specific antibody responses. However, antibodies directed against influenza neuraminidase (NA) also contribute to protection. Here, we analysed the antigenic diversity of a panel of N2 NAs derived from human H3N2 viruses that circulated between 2009 and 2017. The antigenic breadth of these NAs was determined based on the NA inhibition (NAI) of a broad panel of ferret and mouse immune sera that were raised by infection and recombinant N2 NA immunisation. This assessment allowed us to distinguish at least four antigenic groups in the N2 NAs derived from human H3N2 viruses that circulated between 2009 and 2017. Computational analysis further revealed that the amino acid residues in N2 NA that have a major impact on susceptibility to NAI by immune sera are in proximity of the catalytic site. Finally, a machine learning method was developed that allowed to accurately predict the impact of mutations that are present in our N2 NA panel on NAI. These findings have important implications for the renewed interest to develop improved influenza vaccines based on the inclusion of a protective NA antigen formulation.

## eLife assessment

This study presents **valuable** data on the antigenic properties of neuraminidase proteins of human A/H3N2 influenza viruses sampled between 2009 and 2017. The antigenic properties are found to be generally concordant with genetic groups. Compared to a previous version, additional analyses have strengthened the work, with **solid** evidence supporting the claims of the authors.

## Introduction

Neuraminidase (NA) and hemagglutinin (HA) constitute the two major antigenic proteins at the surface of influenza A and B virions (*Uyeki et al., 2022*). While HA mediates binding to sialic acids and membrane fusion, NA catalyses the removal of sialic acid residues from sialylated glycoproteins and glycolipids. This complementary role of HA and NA enables influenza virions to penetrate the mucus barrier that lines the respiratory epithelium and newly formed virions to be released from the infected

**eLife digest** Two proteins, the hemagglutinin and the neuraminidase, protrude from the surface of the influenza virus. Their detection by the immune system allows the host organism to mount defences against the viral threat. The virus evolves in response to this pressure, which manifests as changes in the appearance of its hemagglutinin and neuraminidase. This process, known as antigenic drift, leads to the proteins evading detection. It is also why flu vaccines require frequent updates, as they rely on 'training' the immune system to recognise the most important strains in circulation – primarily by exposing it to appropriate versions of hemagglutinin. While the antigenic drift of hemagglutinin has been extensively studied, much less is known about how the neuraminidase accumulates mutations, and how these affect the immune response.

To investigate this question, Catani et al. selected 43 genetically distant neuraminidases from human viral samples isolated between 2009 and 2017. Statistical analyses were applied to define their relatedness, revealing that a group of closely related neuraminidases predominated from 2009 to 2015, before they were being taken over by a second group. A third group, which was identified in viruses isolated in 2013, was remarkably close to the neuraminidase of strains that circulated in the late 1990s. The fourth and final group of neuraminidases was derived from influenza viruses that normally circulate in pigs but can also occasionally infect humans.

Next, Catani et al. examined the immune response that these 43 neuraminidases could elicit in mice, as well as in ferrets – the animal most traditionally used in influenza research. This allowed them to pinpoint which changes in the neuraminidase sequences were important to escape recognition by the host. Data obtained from the two model species were comparable, suggesting that these experiments could be conducted on mice going forward, which are easier to work with than ferrets. Finally, Catani et al. used machine learning to build a computational model that could predict how strongly the immune system would respond to a specific neuraminidase variant.

These findings could help guide the development of new vaccines that include neuraminidases tailored to best prime and train the immune system against a larger variety of strains. This may aid the development of 'supra-seasonal' vaccines that protect against a broad range of influenza viruses, reducing the need for yearly updates.

cell (*de Vries et al., 2020*; *Matrosovich et al., 2004*; *Palese et al., 1974*). Immunity against influenza is associated with the presence of anti-HA antibodies in serum, as determined by a hemagglutination inhibition assay. In line with this, the composition of licensed influenza vaccines is primarily focused on the antigenic characteristics of HA as the protective vaccine antigen (*Cox, 2013*). Multiple studies, however, have underscored the protective potential of anti-NA immune responses as determined by NA inhibition (NAI) assays (*Couch et al., 2013*; *Maier et al., 2020*; *Memoli et al., 2016*; *Monto and Kendal, 1973*; *Monto et al., 2015*; *Ng et al., 2019*; *Weiss et al., 2020*). There is also evidence that pre-existing anti-N2 immunity induced by prior exposure to H2N2 viruses that circulated in the population from 1957 to 1968 tempered the impact of the H3N2 pandemic virus that emerged in 1968 (*Eickhoff and Meiklejohn, 1969*; *Jester et al., 2020*; *Monto and Kendal, 1973*; *Viboud et al., 2005*). Further, the pharmacological inhibition of NA activity is a well-established therapeutic strategy to try to reduce influenza severity and duration (*Leneva et al., 2000*; *Yen et al., 2005*). In addition, antibodies with NAI activity can protect against influenza virus infection in pre-clinical models (*Chen et al., 2018*; *Deroo et al., 1996*; *Rosu et al., 2022*; *Wan et al., 2018*; *Wohlbold et al., 2017*).

Since its appearance in 1968, human H3N2 viruses have been evolving faster than the co-circulating H1N1 and influenza B viruses (*Bedford et al., 2014*; *Potter et al., 2019*). The overall influenza virus vaccine effectiveness varies between seasons and, despite frequent updates of the H3N2 HA component, the effectiveness of influenza vaccines against H3N2-associated disease ranges between 5% and 39%, which is lower than the effectiveness against H1N1 and influenza B (*Belongia et al., 2016*). In this context the use of NA as a complementary vaccine antigen might improve influenza vaccine effectiveness, e.g., by compensating for HA-based antigenic mismatches in circulating influenza virus strains (*Krammer et al., 2018*).

Phylogenetic analyses of HA and NA genes of human H3N2 viruses revealed the asynchronous evolution of the two proteins. In addition, the evolutionary rate as determined by the accumulation of

mutations over time is estimated to be 1.5 times lower for NA than for HA (*Westgeest et al., 2012*). Despite the strong evidence that immune responses directed against NA can contribute to protection against influenza, a comprehensive insight in the antigenic breadth and evolution of the H3N2 NA antigen is lacking (*Nachbagauer et al., 2017*; *Rajendran et al., 2017*; *Smith et al., 2004*; *Westgeest et al., 2012*).

Here, we determined the antigenic landscape of N2 NAs derived from human H3N2 strains that were isolated between 2009 and 2017. This analysis allowed the identification of at least 4 N2 NA antigenic groups and to estimate the weight of amino acid changes on their contribution to NA antigenic drift.

## Results

### Generation of H3N2-derived NA immune sera in ferrets and mice

Our aim was to determine the antigenic relationship between NAs derived from human H3N2 viruses that circulated during nearly one decade. We first selected 43 H3N2 NA protein sequences obtained via GISAID, derived from viruses that were isolated between 2009 and 2017 (*Elbe and Buckland-Merrett, 2017*). Hierarchical clustering with minimax linkage was employed to ensure the set was representative of overall protein sequence diversity (*Bien and Tibshirani, 2011*). Next, 43 recombinant H1N2 viruses were generated that carried the internal and HA genes from PR8 virus and the NA genes of the selected H3N2 viruses. In addition, the corresponding tetrabrachion-stabilised recombinant N2 NA proteins were produced in Chinese hamster ovary (CHO) cells and purified, and are referred to as NACHOs (*Catani et al., 2022*). All purified NACHOs were tetrameric based on SEC-MALS analysis. The enzymatic activity of the NACHOs was determined based on the hydrolysis of the fluorogenic small molecule substrate 4-methylumbelliferyl-*N*-acetylneuraminic acid (MUNANA). The NACHOs were active with a specific activity ranging from 0.27 to 27.48 nmole/min/µg, indicating correct folding and assembly of the NA head domains (*Supplementary file 1*).

Immune sera against all 43 H3N2-derived NAs were generated in pairs of ferrets by a primary infection with the respective H1N2 reassortant viruses. This prime by infection was used to avoid the use of an adjuvant upon subsequent boost with the corresponding NACHOs. On day 28 after primary infection, all animals were boosted by intramuscular injection of homologous NACHO, except for the ferrets that had been primed by infection with A/Stockholm/15/2014 and A/Estonia/91621/2015, which were re-inoculated with the same respective H1N2 reassortant viruses. These two pairs of ferrets were not boosted with homologous NACHO because the corresponding NACHOs were not available at the time of the experiment. Blood samples were taken from the ferrets on days 14 and 42 after (primary) infection (*Figure 1a*; *Figure 1—source data 1*). Priming by H1N2 infection resulted in seroconversion against the respective immobilised homologous NACHOs (*Figure 1b*; *Figure 1—source data 1*). Subsequent immunisation with recombinant NACHOs boosted the NACHO-specific IgG titers in day 42 serum samples. In contrast, serum samples from the ferrets that had been re-inoculated with H1N2 reassortant carrying NA derived from A/Stockholm/15/2014 or A/Estonia/91621/2015 showed no further increase in titers, presumably because prior exposure with the same H1N2 virus had induced a sterilising immune response (*Figure 1b*).

We also generated NA immune sera in BALB/c mice. Groups of five mice were immunised two times, with a 2-week interval, by intramuscular injection with 1 µg of the respective 43 NACHOs adjuvanted with AF03. Blood was collected from the immunised mice and serum prepared on day 28 after the prime immunisation (*Figure 1c*). NACHO-specific IgG titers in the mouse immune sera, pooled from each group, ranged from 1:8100 to 1:72,900 (*Figure 1d*).

### The antigenic profile of N2 NA

Next, NAI titers were determined in the ferret sera using a panel of H6N2 reassortant viruses in an enzyme-linked lectin assay (ELLA), avoiding interference by anti-H1 HA antibodies. We generated 27 H6N2 influenza viruses with H6 HA derived from A/mallard/Sweden/2002 and NA derived from a subset of the 43 H3N2 viruses, thereby retaining NA sequence diversity and year of H3N2 virus isolation (*Supplementary file 2*).

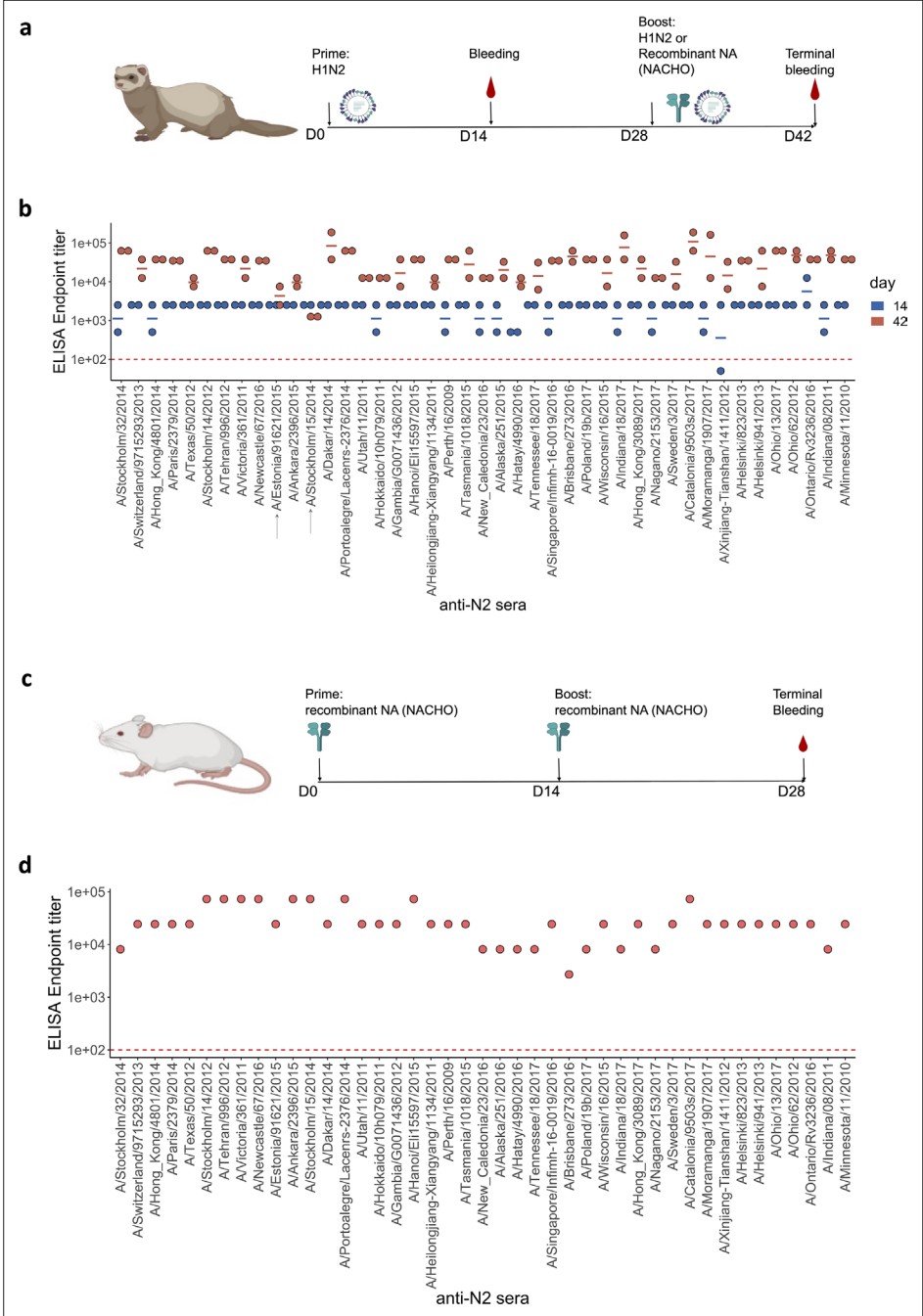

**Figure 1.** Generation of anti-N2 sera panel from ferrets and mice. (**a**) 43 Ferret immunisation and blood sampling scheme. The second immunisation on day 28 was performed with homologous NACHO except for the ferrets that had been primed with H1N2$_{A/Estonia/91621/2015}$ or H1N2$_{A/Stockholm/15/2014}$, which were infected again on day 28 post prime with the same H1N2 viruses. (**b**) Homologous NACHO ELISA titer of ferret sera sampled on day 14 after inoculation with H1N2 virus and 2 weeks after the boost with recombinant neuraminidase (NA) or by re-inoculation with H1N2$_{A/Estonia/91621/2015}$ or H1N2$_{A/Stockholm/15/2014}$ (indicated by arrows on the x-axis). (**c**) 43 BALB/c mouse immunisation and blood sampling scheme. (**d**) Homologous NACHO serum IgG titers in pooled sera from mice that were immunised twice with recombinant NA. Mouse and ferrets representations were created with BioRender.com, and published using a CC BY-NC-ND license with permission.

The online version of this article includes the following source data for figure 1:

**Source data 1.** Total IgG ELISA titers against homologous recombinant N2 neuraminidases.

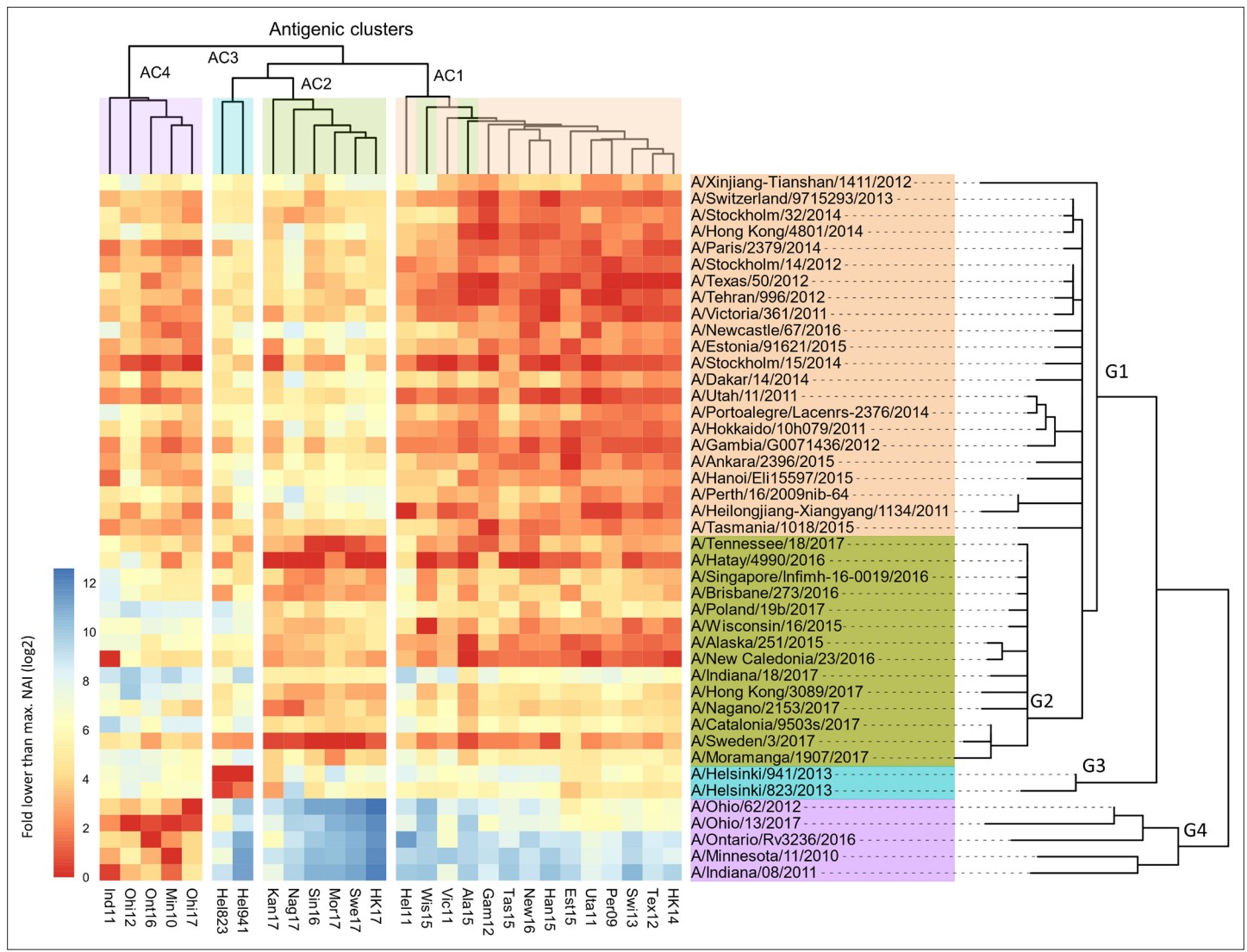

**Figure 2.** Breadth of N2 neuraminidase (NA) inhibition. 43 Phylogenetic tree based on the N2 NA head domain amino acid sequences and heatmap representing the average of normalised NA inhibition (NAI) per H6N2 [$\log_2$ max (NAI/ELISA homologous titer)/(NAI/homologous ELISA titer)] determined in ferret sera after the boost (listed vertically). The red-to-blue scale indicates high-to-low NAI observed in enzyme-linked lectin assay (ELLA) against the H6N2 reassortants (listed at the bottom). Unweighted pair group method with arithmetic mean (UPGMA) clustering of H6N2s inhibition profiles are shown on top of the heatmap and coloured according to the phylogenetic groups.

The online version of this article includes the following source data, source code, and figure supplement(s) for figure 2:

**Source data 1.** Neuraminidase inhibition of H6N$_x$ strains by ferrets sera.

**Source data 2.** Neuraminidase inhibition of H6N$_x$ strains by mouse sera.

**Figure supplement 1.** The N2 neuraminidase (NA) substrate affinity is distinct in the N2 phylogenetic groups.

**Figure supplement 2.** Antigenic relationship between neuraminidase (NA) of the H6N2 viruses.

**Figure supplement 3.** Antigenic map of N2 neuraminidases (NAs).

**Figure supplement 3—source code 1.** Antigenic map generation and quality control.

**Figure supplement 4.** Correlation of neuraminidase inhibition (NAI) fold reduction and phylogenetic relatedness.

**Figure supplement 5.** Breadth of N2 inhibition of sera from ferrets primed by H1N2 infection only.

**Figure supplement 6.** Breadth of N2 inhibition is confirmed in mouse sera.

The 43 NAs used as antigens and 27 NAs as source of NA could be classified into four groups when ordered according to the phylogenetic relatedness of their NA sequence (*Figure 2*; *Figure 2—source data 1*).

The conversion rate of MUNANA into 4-methylumbelliferyl and acetylneuraminic acid by these H6N2 viruses revealed a $K_M$ that ranged from 20.7 to 144.8 µM as determined by the Michaelis-Menten equation (*Figure 2—figure supplement 1a* and *Supplementary file 2*). Interestingly, the $K_M$ differs between the distinct phylogenetic groups (*Figure 2—figure supplement 1a*). There was a negative correlation between the $K_M$ values and the dilution of the H6N2 viruses required to set the 70% maximal NA activity in ELLA (*Figure 2—figure supplement 1b*).

Using this panel of 27 H6N2s, we determined the NAI titers in the sera obtained from the 43 pairs of ferrets 2 weeks after the boost. ELLAs were performed in duplicate for all serum samples against one H6N2 target at the same time. The reciprocal NAI titers of the ferret sera paired with H6N2 viruses are shown in *Supplementary file 3*. In general, the NAI titers of the ferret sera were the highest against homologous and phylogenetically closely related H6N2 viruses.

In *Figure 2*, all the tested ferret sera are represented in the rows and the 27 H6N2 viruses against which the sera were evaluated in ELLA are presented in the columns. The serum samples were ordered according to the phylogenetic relatedness of the N2 NA head domain amino acid sequences, showing that these cluster in four groups (G1-G4). The ELLA-based $IC_{50}$ NAI titers were normalised by the homologous ELISA titers and then plotted relative to the maximum NAI titer obtained against the respective H6N2 target viruses. The heatmap thus represents, for each H6N2 virus, the normalised ELLA $IC_{50}$ titer expressed as $\log_2$ of the fold-difference relative to the serum with the highest $IC_{50}$ value against that H6N2 virus (dark red squares indicate high and blue squares low $IC_{50}$ values) (*Figure 2*).

Next, these normalised NAI values were clustered against each of the H6N2s using the unweighted pair group method with arithmetic mean (UPGMA), which is a distance-based method. This clustering allowed to discern at least four major antigenic clusters (AC1 to –4) that are depicted at the top of the heatmap (*Figure 2*). It is interesting to note that the H6N2 viruses with NA antigens derived from A/Wisconsin/16/15 and A/Alaska/251/15 do not segregate into AC2, as would be predicted based on the phylogenetic relatedness, but cluster in AC1 based on the UPGMA. However, the overall concordance of the antigenicity with the phylogenetic relatedness, based on the amino acid sequences of the NA head domains, was confirmed by hierarchical clustering with bootstrapping and *k*-means (*Figure 2—figure supplement 2a and b*).

In addition, antigenic cartography analysis, based on the method described by *Smith et al., 2004*, resulted in an antigenic map of the N2 NAs that was in line with the four proposed antigenic clusters (*Figure 2—figure supplement 3*).

In line with this, a positive correlation was observed between the pairwise phylogenetic distances and the normalised NAI $IC_{50}$ values for each of the tested H6N2s (*Figure 2—figure supplement 4*).

We further tested the NAI activity of ferrets' sera obtained 2 weeks after the prime infection against at least one H6N2 virus from each antigenic cluster. These H6N2 viruses were selected based on their antigenic relatedness as deduced from the ferret immune sera analysis (*Figure 2*) with two representatives from group 1 (Tex12 and Per09) and one each from group 2 (Kan17), group 3 (Hel823), and group 4 (Ind11). The normalised data confirmed the antigenic clustering observed in sera after boost (*Figure 2—figure supplement 5*).

The NAI titers were also determined in the sera obtained from the 43 groups of NACHO-immunised mice against 7 of the H6N2 viruses (*Figure 2—figure supplement 6a*; *Figure 2—source data 2*). These H6N2 viruses were selected as described above with the addition of HK14 and Swi13 H6N2s, both belonging to antigenic cluster 1. Serum NAI $IC_{50}$ values against a given H6N2 virus were normalised against the $IC_{50}$ of the mouse serum that had the highest NAI titers against the tested H6N2 virus. UPGMA-based clustering showed that the mouse NACHO immune sera displayed a similar recognition pattern as the ferret immune sera that had been generated by H1N2 priming followed by a NACHO boost (*Figure 2*). For example, HK14 H6N2 NA activity is strongly inhibited by sera from group 1, is poorly inhibited by most of the sera from group 2, and is hardly inhibited by group 3 and 4 sera from both ferrets and mice. Ind11 H6N2 NA activity, on the other hand, was inhibited by group 1 and 4, and hardly by group 2 and 3 immune sera. Overall, the normalised NAI $IC_{50}$ values of the ferret and mouse immune sera correlated very well with coefficients values ranging from 0.33 to 0.78 (*Figure 2—figure supplement 6b*).

## Association analysis reveals amino acid residue substitutions that strongly impact NAI

NA of human influenza viruses is subject to antigenic drift, predominantly driven by the NAI activity that is detectable in the host serum (*Air et al., 1985*; *Yasuhara et al., 2019*). There are a total of 102 amino acid differences between the head domains of the NAs that were used to generate the 43 ferret immune sera (*Figure 3—figure supplements 1 and 2*).

To try to identify which of these amino acid substitutions have a significant impact on NAI, we performed an association analysis for each of the tested H6N2 viruses. For this analysis, the NAI $IC_{50}$ value was compared between groups of N2 sera according to the presence or absence of a substitution of any of the 102 amino acid position that are different between the panel of 43 tested N2s against which sera had been raised in ferrets. As a result, a p-value is obtained for each position that reflects its possible impact on NAI (*Figure 3a*). The significance values higher than 1.3 (p<0.05) of representative strains of the four antigenic groups were represented on the surface of N2 (*Figure 3b*). No obvious antigenic patches, however, could be distinguished on the N2 NA surface.

A similar association analysis was performed using the NAI values obtained with the mouse NACHO immune sera. The calculated p-values obtained from the mouse and ferret NAI data were in line with each other (*Figure 3—figure supplement 3*).

## Residues close to the catalytic pocket are more likely to impact NAI

We next calculated the distance between the Cα atom of each amino acid present in the N2 structure (PDB: 6BR6, from A/Perth/16/2009 NA) to each Cα of the variable amino acids present in the panel. We then determined the correlation of those distances with the p-values that were previously determined in the association analysis. The resulting correlation coefficients (Tau values) indicate a similar pattern among all 27 tested H6N2s (*Figure 4a*). When visualised on the N2 surface structure, the Tau values revealed a geometrical distribution with the negative correlations being in closer proximity to the catalytic pocket (*Figure 4b*). Similar data was obtained when mice sera were analysed this way (*Figure 4—figure supplement 1*).

## Prediction of the N2 antigenicity based on a machine learning method

The antigenic drift of human H3 HA has been extensively studied and modelled (*Meyer and Wilke, 2015*; *Smith et al., 2004*; *Westgeest et al., 2012*; *Wu et al., 2020*). To model the NA drift observed in our panel of N2 NAs, we first calculated the antigenic distances between the distinct NA sera, using a similar approach as has been used to visualise the HA antigenic map (*Cai et al., 2012*). Then, using the antigenic distances and a random forest (RF) method, a model was built based on amino acid sequence information. A similar model has been used by Yao et al. to predict the antigenicity of H3 HA (*Yao et al., 2017*). Here, the optimised model was built using a combination of 25 randomly sampled features in 1500 decision trees. The resulting model generates predictive antigenic distances that strongly correlate with the observed values (*Figure 5a*), with a root-mean-square error (RMSE)/mean = 0.207. The RF also enables scoring of amino acid residues according to their importance in the decision trees. One measure of importance can be defined as how early a change in a position is placed in the generated decision trees (mean minimal depth). Another measure of importance is how much the standard error mean increases when a variable is permuted. Finally, the prevalence of a variable amino acid in nodes of the decision tree can also indicate its importance. Those importance scores were represented in a multi-way plot (*Figure 5b*). The importance scores highlighted similar positions as previously determined in the association study. Finally, we used NA sequences of recent seasonal influenza vaccine recommended H3N2 strains to predict their antigenic distance relative to our panel of N2s, including three recent N2 NA components (from A/HongKong/45/2019, A/Tasmania/503/2020, and A/Darwin/6/2021) that were not present in our dataset. The predicted antigenic distances were plotted in the heatmap in a red-to-blue scale, indicating high to low antigenic distance, respectively (*Figure 5c*). The RF clearly reproduced the drift observed in AC2 versus AC1. The modelling also indicates that the antigenic distance of the three recent N2s to their corresponding group 2 NAs is increasing (represented by the dark blue turning light blue, from the A/Singapore/infimh-16-0019/2016 towards the A/Darwin/6/2021 in G2) whereas their distance to group 4 NAs is decreasing (dark red becoming yellow from the A/Singapore/infimh-16-0019/2016 towards the A/Darwin/6/2021 in G4).

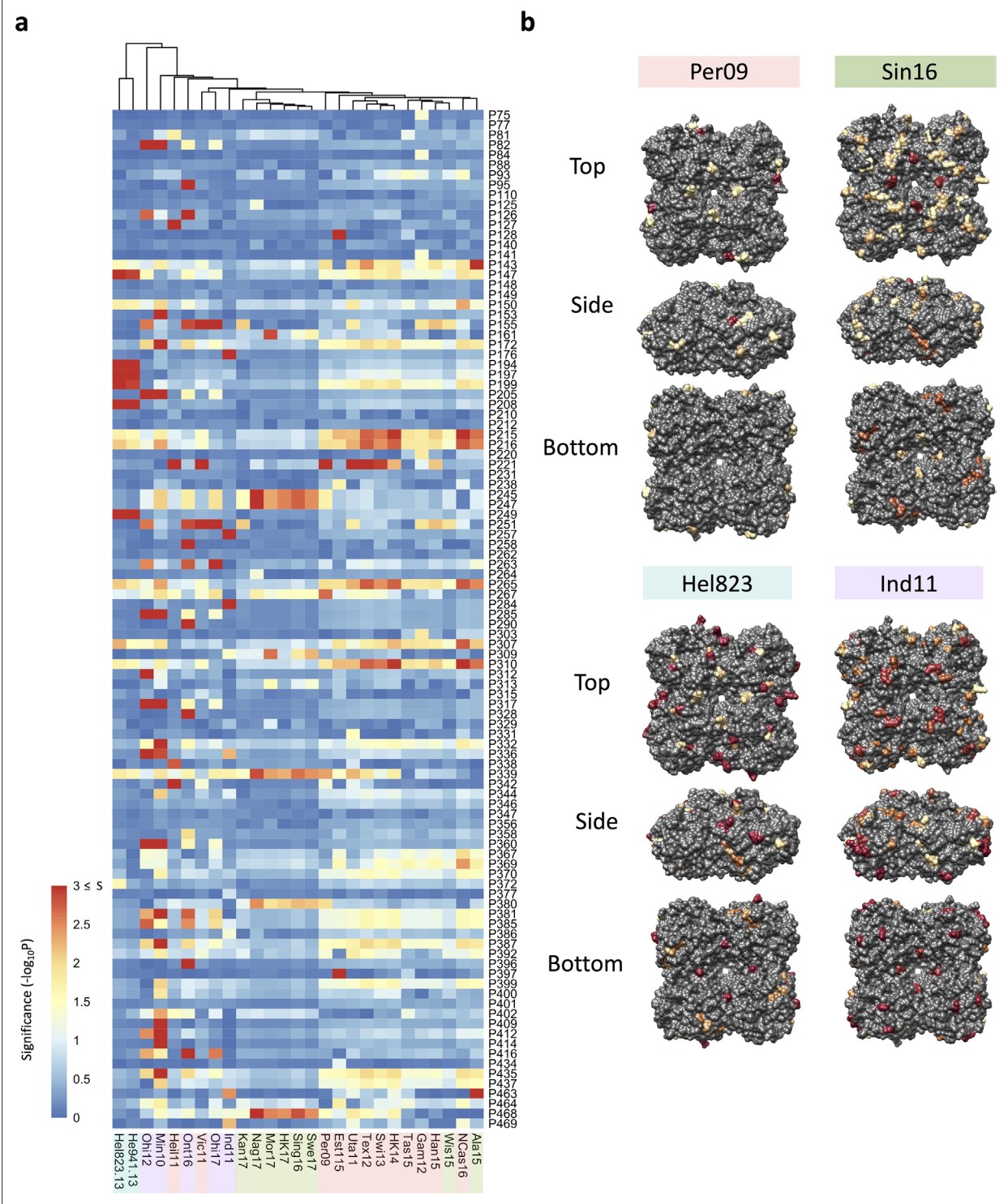

**Figure 3.** Association analysis reveals impact of amino acid mutations on neuraminidase inhibition (NAI). (**a**) Association study performed for each H6N2 virus (listed at the bottom) to determine the probability of each variable amino acid (listed on the right) present in the panel of 43 NAs to impact the NAI. The red-to-blue scale represents significance values ($S=-\log_{10}p$) associated with each amino acid substitution. The dark red in the colour scale represent all values that are equal or higher than 3 (corresponding to p-values ≤ 0.001). The colour shades on the bottom side indicate the distinct phylogenetic groups. (**b**) The significant values (p<0.05) were represented on the surface of N2 tetramer (PDB accession number 4H53) for one representative strain of each phylogenetic group with a yellow to red scale of significance ($-\log_{10}p$).

The online version of this article includes the following figure supplement(s) for figure 3:

*Figure 3 continued on next page*

**Figure supplement 1.** Protein sequence alignment of N2 NACHO panel used for immunisation.

**Figure supplement 2.** Geometrical distribution of amino acid substitutions in the N2 structure.

**Figure supplement 3.** Association study was performed on data obtained from mouse sera.

## Discussion

The sequence diversity of N2 proteins derived from the 43 selected human H3N2 viruses isolated between 2009 and 2017 allows to classify them in four major phylogenetic groups. Group 1 contains 22 N2 NAs, 4 of which were derived from vaccine recommended H3N2 strains: A/Perth/16/2009, A/

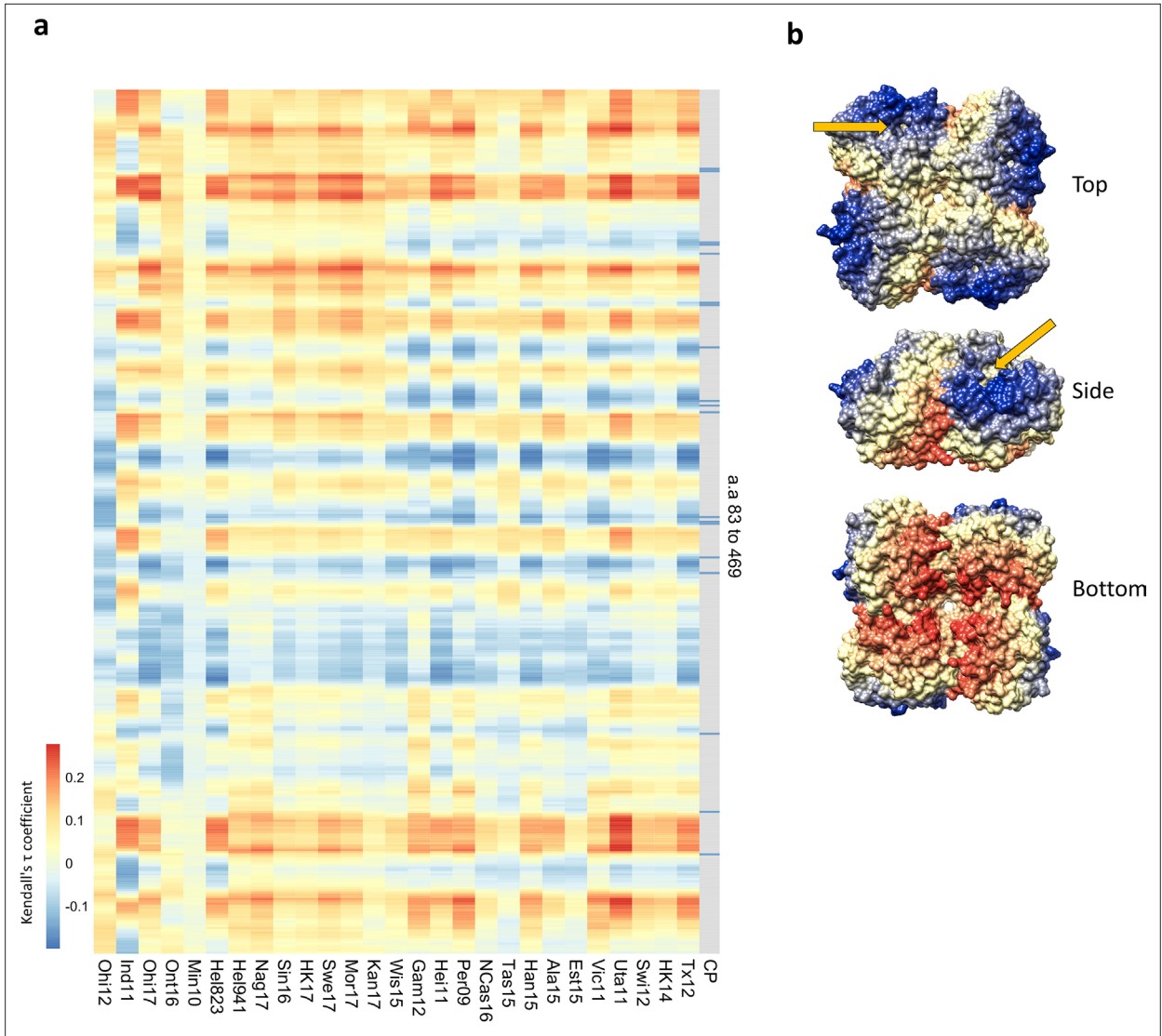

**Figure 4.** Residues close to the catalytic pocket are more likely to impact neuraminidase inhibition (NAI). (**a**) Significance values (102 variable amino acids) obtained from the association study were correlated to the distance between every amino acid and each of the variable amino acids in the NA structure. The red-to-blue scale represents the Tau values obtained in the Kendall correlation. Amino acids 83–469 are shown from top to bottom in the heatmap. The H6N2s used in the NAI panel are listed at the bottom. The grey column indicates the residues that are part of the catalytic pocket (CP) that directly interact with the substrate (R118, D151, R152, R224, E276, R292, R371, and Y406) and the residues that play a key role in stabilising the catalytic site (E119, R156, W178, S179, D198, I222, E227, H274, E277, N294, and E425). (**b**) Tau values obtained from Tex12 H6N2 correlation are represented on the surface of the N2 structure of A/RI/5+/1957 (PBD 4H53), the arrow indicates the catalytic pocket in one NA protomer.

The online version of this article includes the following figure supplement(s) for figure 4:

**Figure supplement 1.** Correlation of values obtained from the mouse sera panel and amino acid distances.

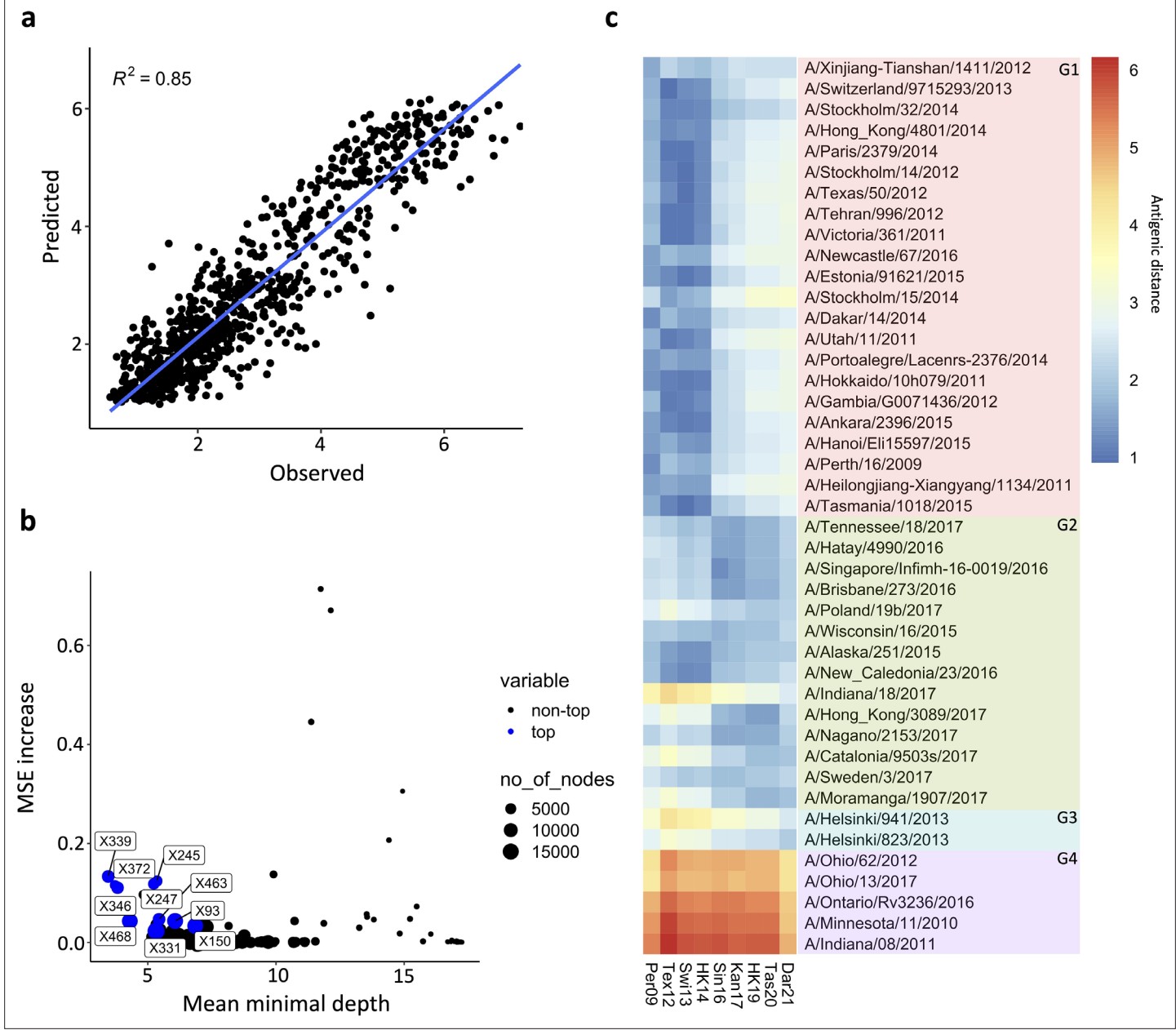

**Figure 5.** Prediction of N2 antigenicity based on machine learning. (**a**) Predicted vs observed normalised neuraminidase inhibitions (NAIs) are shown. (**b**) Multi-way importance plot indicating the increase in mean standard error, the mean depth, and number of nodes. The top 10 most important amino acid changes are highlighted in blue and annotated with their position in the NA sequence. (**c**) 43 Heatmap indicates the predicted antigenic distances of N2s from recent H3N2 isolates (at the bottom of the heatmap) relative to the N2s that were used in the NAI breadth panel (listed vertically). The antigenic distances were calculated based on NA protein sequence.

The online version of this article includes the following source code for figure 5:

**Source code 1.** Random forest modeling.

Texas/50/2012, A/Switzerland/9715293/2013, and A/Hong Kong/4801/2014. Phylogenetic group 2 comprises 14 N2 NAs, 1 of which is derived from a vaccine H3N2 recommended strain: A/Singapore/INFIMH-16-0019/2016. The third phylogenetic group contains two NAs that are derived from H3N2 strains that had been isolated in Helsinki in 2013. Remarkably, the NA sequence of these two strains is closely related to that of H3N2s that circulated more than 15 years earlier (e.g. 99% identity with NA of A/Sydney/5/1997). The fourth phylogenetic group is composed of NAs derived from zoonotic cases with swine H3N2 strains isolated between 2012 and 2017. Swine-to-human transmission of H3N2

viruses have occurred multiple times in the USA in 2012 and continue to be detected at low frequency until to date (*CDC, 2023*; *Elbe and Buckland-Merrett, 2017*).

We assessed the antigenic diversity of the NA panel by generating 27 recombinant H6N2 viruses and 43 ferret NA immune sera produced by H1N2 priming followed by boosting with homologous recombinant NA (NACHO; *n*=41) or by reinfection with H1N2 (A/Estonia/91621/2015 and A/Stockholm/15/2014). Most of the ferrets had detectable serum anti-N2 antibodies by day 14 after infection with H1N2. Anti-NA antibody titers increased after a boost with non-adjuvanted homologous recombinant CHO-produced soluble tetrameric NA suggesting that the use of recombinant NA may boost naturally acquired anti-NA antibody responses in the human population.

The NAI activity of the ferret sera against 27 tested H6N2s reassortants revealed a pattern of inhibition that largely coincides with the phylogenetic distances based on the NA head domain sequences. In the tested panel of H6N2s, at least four major antigenic groups were identified. We noted that H6N2 with Kan17 NA had a broader profile of inhibition than other H6N2 viruses from group 2, being also inhibited by serum raised against NA from phylogenetic groups 3 and, to a lesser extent, group 4. Additionally, the H6N2s Wis15 and Ala15 have an antigenic behaviour that is more closely related to antigenic cluster 1, despite belonging to group 2 according to their phylogenetic similarities. Based on the NAI panel, an association analysis was performed to try to identify amino acid changes that affect NAI. It is interesting to note that the amino acid substitutions S245N and S247T, combined with substitution P468H, were identified in this association analysis. Those substitutions were described as major contributors of the NA antigenic drift that occurred in H3N2 in 2014 (*Wan et al., 2019*). NAs from A/Wisconsin/16/2015 and A/Alaska/251/2015 also have the S245N and S247T substitutions and phylogenetically belong to group 2 NAs. However, those 2 NAs lack the P468H mutation and are antigenically closer to group 1 NAs, confirming the requirement of P468H combined with the S245N and S247T substitutions for this NA antigenic drift (*Wan et al., 2019*). This observation suggests that NA antigenic drift may sometimes require a combination of amino acid changes rather than single mutations. Wang and colleagues have very elegantly demonstrated that N2 antigenic evolution is constrained by charge balances in an antigenic region, which can partially explain the observation that NA amino acids can undergo co-evolution to result in antigenic drift (*Wang et al., 2021*). Interestingly, NAs originated from H3N2 variants classified in phylogenetic group 4 were antigenically closer to N2s belonging to phylogenetic group 1. This suggest that the antigenic drift of recent NAs from group 2 increases the antigenic distance from the N2s of swine origin.

Noteworthy, the analysis of the sera obtained from ferrets after infection shows that, despite the lower homologous anti-NA titers, the inhibition pattern evaluated against five distinct H6N2s (composed of at least one representative of each antigenic group) is in accordance with the response obtained after boosting with recombinant protein. This suggests that a boost with homologous recombinant soluble tetrameric NA does not alter the breadth of the antibody response induced by H1N2 infection only. Nevertheless, we cannot exclude a degree of increased cross-reactivity of the sera from NACHO-boosted ferret. It has been reported that repeated H3N2 virus exposure of ferrets affects anti-HA responses (*Kosikova et al., 2018*).

NAI was also determined against 7 distinct H6N2 reassortant viruses using a panel of 43 anti-N2 NA sera raised in mice by immunisation with adjuvanted NACHOs. The observed NAI responses again allowed clustering of the N2 NAs into four main antigenic groups, which correlates with data obtained in ferrets. Studies in mice can thus offer a convenient alternative to the use of ferrets in future NA antigenicity studies. This data also supports the hypothesis that there may be universal rules of immunogen antigenicity and immunogenicity (*Altman et al., 2015*).

The determined $K_M$ values of the H6N2 viruses used here indicate that substrate affinity tends to vary according to the phylogenetic relatedness of N2s. The N-glycosylation at position 245 was previously described to be responsible for reducing N2 affinity for its substrate (*Powell and Pekosz, 2020*). Unlike group 2, which is primarily characterised by the presence of an N-glycan at position 245, NAs from phylogenetic group 4 have a low $K_M$, indicating an increased affinity for substrate. This result suggests that phylogenetic drift can be accompanied by changes in NA enzymatic properties. Curiously, the Ala15, Wis15, and Kan17, despite having antigenic properties closer to group 1, still have $K_M$ values that are comparable to those of group 2 N2 NAs.

After the determination of NA antigenic distances, we used RF modelling to estimate the impact of amino acid changes on antigenic distances. We obtained a strong correlation of predicted versus

observed antigenic distances. However, it is important to note that the training dataset was based on the 102 mutations present in our panel, limiting the predictions for other positions. Furthermore, RF enabled the classification of the variable amino acids according to importance scores. Also, our modelling does not consider that substitution by other amino acids can have a distinct impact on the antigenic distance. As a consequence, predictions based on the model could underestimate the importance of a particular amino acid residue substitution in some cases. It is important to highlight that co-occurring substitutions in our panel (the ones present in the main branches of the phylogenetic tree) cannot be individually assessed by association analysis or the RF model. The individual weight of those mutation on NA drift thus remains to be experimentally demonstrated. Interestingly, the modelling could put in evidence the gradual drift of recent N2s which were not present in the dataset (A/Hong Kong/45/2019, A/Tasmania/503/2020, and A/Darwin/6/2021).

It has been reported that an N-glycosylation site at position 329 combined with E344 in NA from human H3N2 viruses from 2007 to 2013 was gradually lost in later H3N2 viruses (*Ge et al., 2022*). This loss of an N-glycosylation site at position 329 combined with an E344K substitution was associated with a change in NAI reactivity in ferret sera. Three N2 NAs in our panel, derived from A/Victoria/361/2011, A/Hong_Kong/3089/2017, and A/Tennessee/18/2017, lack this N-glycosylation motif. The E344K substitution is present in three other NAs, derived from A/Nagano/2153/2017, A/Minnesota/11/2010, and A/Indiana/08/2011. The importance of those mutations is among the lowest ones predicted by our modelling. However, the differences in NAI reported by Ge et al. are very modest (less than twofold). The experimental variability in our study potentially limits the identification of substitutions with a subtle impact on NAI.

The protective potential of NA antigen against influenza has been underscored by multiple studies. It is likely that NA will be included as a standard component of next-generation influenza vaccines. The difficulties of producing stable tetrameric NA may be overcome in the future, thanks to protein design or with the advent of mRNA-based vaccines (*Catani et al., 2022*; *Chivukula et al., 2021*; *Ellis et al., 2022*). The antigenic modelling associated with sequence surveillance may increase our predictive ability, highlighting major potential drifts and support NA antigen designs that may elicit broader NAI responses compared to natural NA.

In summary, our study shows that immunisation with recombinant N2 NA immunogen can boost anti-NA serum antibody responses in ferrets that were primed by infection with an influenza virus that carries the homologous NA. At least four major antigenic groups could be distinguished in N2 NA derived from human H3N2 viruses that were isolated between 2009 and 2017, and those antigenic groups predominantly correspond with the phylogenetic relatedness of the N2 head domains. In addition, the amino acid substitutions in close proximity of the NA catalytic pocket are more likely to affect NAI patterns.

# Materials and methods

## Key resources table

| Reagent type (species) or resource | Designation | Source or reference | Identifiers | Additional information |
|---|---|---|---|---|
| Gene (influenza A) | NA | GISAID | EPI232655 | A/PERTH/16/2009 |
| Gene (influenza A) | NA | GISAID | EPI543762 | A/SWITZERLAND/97 15293/2013 |
| Gene (influenza A) | NA | GISAID | EPI993642 | A/TENNESSEE/18/2017 |
| Gene (influenza A) | NA | GISAID | EPI715819 | A/ANKARA/2396/2015 |
| Gene (influenza A) | NA | GISAID | EPI574238 | A/PARIS/2379/2014 |
| Gene (influenza A) | NA | GISAID | EPI369152 | A/STOCKHOLM/14/2012 |
| Gene (influenza A) | NA | GISAID | EPI892776 | A/BRISBANE/273/2016 |

*Continued on next page*

*Continued*

| Reagent type (species) or resource | Designation | Source or reference | Identifiers | Additional information |
|---|---|---|---|---|
| Gene (influenza A) | NA | GISAID | EPI1013098 | A/HATAY/4990/2016 |
| Gene (influenza A) | NA | GISAID | EPI1015697 | A/CATALONIA/9503S/2017 |
| Gene (influenza A) | NA | GISAID | EPI531449 | A/STOCKHOLM/15/2014 |
| Gene (influenza A) | NA | GISAID | EPI679413 | A/ALASKA/251/2015 |
| Gene (influenza A) | NA | GISAID | EPI630701 | A/ESTONIA/91621/2015 |
| Gene (influenza A) | NA | GISAID | EPI394905 | A/OHIO/62/2012 |
| Gene (influenza A) | NA | GISAID | EPI446773 | A/TEHRAN/996/2012 |
| Gene (influenza A) | NA | GISAID | EPI831629 | A/NEWCASTLE/67/2016 |
| Gene (influenza A) | NA | GISAID | EPI831647 | A/NEWCALEDONIA/23/2016 |
| Gene (influenza A) | NA | GISAID | EPI935704 | A/WISCONSIN/16/2015 |
| Gene (influenza A) | NA | GISAID | EPI981927 | A/INDIANA/18/2017 |
| Gene (influenza A) | NA | GISAID | EPI1055123 | A/NAGANO/2153/2017 |
| Gene (influenza A) | NA | GISAID | EPI408590 | A/XINJIANGTIANSHAN/1411/2012 |
| Gene (influenza A) | NA | GISAID | EPI330528 | A/HEILONGJIANGXIANGYANG/1134/2011 |
| Gene (influenza A) | NA | GISAID | EPI316206 | A/UTAH/11/2011 |
| Gene (influenza A) | NA | GISAID | EPI536339 | A/GAMBIA/G0071436/2012 |
| Gene (influenza A) | NA | GISAID | EPI1036140 | A/POLAND/19B/2017 |
| Gene (influenza A) | NA | GISAID | EPI675974 | A/TASMANIA/1018/2015 |
| Gene (influenza A) | NA | GISAID | EPI539797 | A/DAKAR/14/2014 |
| Gene (influenza A) | NA | GISAID | EPI340631 | A/HOKKAIDO/10H079/2011 |
| Gene (influenza A) | NA | GISAID | EPI465399 | A/MINNESOTA/11/2010 |
| Gene (influenza A) | NA | GISAID | EPI956450 | A/HANOI/ELI15597/2015 |
| Gene (influenza A) | NA | GISAID | EPI869121 | A/ONTARIO/RV3236/2016 |
| Gene (influenza A) | NA | GISAID | EPI344404 | A/INDIANA/08/2011 |
| Gene (influenza A) | NA | GISAID | EPI558926 | A/STOCKHOLM/32/2014 |
| Gene (influenza A) | NA | GISAID | EPI104388 | A/HONGKONG/3089/2017 |

*Continued*

| Reagent type (species) or resource | Designation | Source or reference | Identifiers | Additional information |
|---|---|---|---|---|
| Gene (influenza A) | NA | GISAID | EPI1056652 | A/OHIO/13/2017 |
| Gene (influenza A) | NA | GISAID | EPI489603 | A/HELSINKI/941/2013 |
| Gene (influenza A) | NA | GISAID | EPI489571 | A/HELSINKI/823/2013 |
| Gene (influenza A) | NA | GISAID | EPI1050862 | A/MORAMANGA/1907/2017 |
| Gene (influenza A) | NA | GISAID | EPI996522 | A/PORTOALEGRE/LACENRS2376/2014 |
| Gene (influenza A) | NA | GISAID | EPI945994 | A/SWEDEN/3/2017 |
| Gene (influenza A) | NA | GISAID | EPI1047603 | A/SINGAPORE/INFIMH160019/2016 |
| Gene (influenza A) | NA | GISAID | EPI377498 | A/TEXAS/50/2012 |
| Gene (influenza A) | NA | GISAID | EPI1026710 | A/HONGKONG/4801/2014 |
| Gene (influenza A) | NA | GISAID | EPI551805 | A/VICTORIA/361/2011 |
| Gene (influenza A) | HA | GISAID | EPI251808 | A/mallard/Sweden/81/2002 |
| Cell line (*Canis lupus*) | MDCK | ATCC | CCL-34 | |
| Cell line (*Homo sapiens*) | 293FT | Thermo Fisher Scientific | R70007 | |
| Biological sample (*Mus musculus*) | Sera | This paper | | Harvested from immunised mice |
| Biological sample (*Mustela furo*) | Sera | This paper | | Harvested from infected/immunised ferrets |
| Antibody | Anti-mouse IgG-HRP (Goat polyclonal) | GE Healthcare | NA931-1ml | (1:5000) |
| Antibody | Anti-ferret IgG-HRP (Goat polyclonal) | Abcam | 112770 | (1:100,000) |
| Chemical compound, drug | 3,3',5,5'-Tetramethylbenzidine | BD | 555214 | |
| Chemical compound, drug | Fetuin | Sigma | F3385 | |
| Chemical compound, drug | PNA-HRP | Sigma | L61135-1MG | |
| Chemical compound, drug | 4-Methylumbelliferyl-α-d-N-acetylneuraminic acid | Sigma | M8639 | |
| Other | Pierce nickel-coated plates | Thermo Fisher Scientific | 15442 | See Materials and methods: ELISA |

## Rescue of H1N2 and H6N2 reassortants

All reassortant H1N2 and H6N2 viruses included in the study were generated by reverse genetics, with each reassortant expressing the targeted NA antigen, internal genes from A/Puerto Rico/8/1934 H1N1 (PR8), and the HA from PR8 H1N1 or A/mallard/Sweden/81/2002 H6N1, respectively. Briefly, HA and NA segments including non-coding regions were generated by custom gene synthesis (Geneart AG), while PR8 internal segments were derived from a viral isolate as previously published (*Marsh and Tannock, 2005*). All segments were cloned into a bidirectional transcription plasmid derived from pUC57 (Genscript) including polymerase (Pol) I and Pol II promoters, as previously described (*Hoffmann et al., 2000*), and the entire set of eight plasmids was used to transfect 293FT cells (Thermo Fisher Scientific) using Lipofectamine 2000 CD (Thermo Fisher Scientific). Twenty-four hours after transfection, Madin-Darby Canine Kidney Cells (MDCK-) ATL cells (ATCC) were added to the transfected cells in the presence of TPCK-treated trypsin (Sigma) to allow influenza virus propagation. Cell culture supernatants containing influenza virus were harvested 7 days post MDCK addition and blindly

passaged in 8- to 10-day-old embryonated chicken eggs (Charles River Laboratories, Inc). Inoculated eggs were incubated at 37°C for 48 hr, then cooled to 4°C for 12 hr, prior to allantoic fluid harvest and clarification by low-speed centrifugation (3000 rpm, 20 min). For H6N2 viruses only, high-yield stocks were generated by an additional passage in eggs as described above. Virus titers were determined by plaque assay on MDCK cells.

## Design and production of recombinant proteins

The coding information of the NACHO proteins was cloned under the transcriptional control of the CMV promoter in the pCDNA3.4 plasmid with a CD5 secretion signal, an amino-terminal His-tag and the thrombin cleavage signal followed by tetrabrachion-NA. NACHOs were expressed in a mammalian cell culture system as previously described with modifications (*Catani et al., 2022*).

## Animal experiments

The ferret immunisation experiment was conducted at Bioqual, Inc, MD (USA). The experiments were approved by the Institutional Animal Care and Use Committee of the Bioqual (reference number 17-021.6). Outbred 18- to 22-week-old naïve male and female Fitch ferrets were obtained from Triple F, PA (USA) and housed under specified pathogen-free conditions with food and water ad libitum. Ferrets were primed on day 0 by intranasal challenge with influenza H1N2 reassortant virus (1000 µl/ dose, split evenly between nostrils). Four weeks after initial infection (day 28), ferrets were immunised with 45 µg of NACHO (500 µl/dose, intramuscularly, split evenly between hind thighs) corresponding to homologous NA used in day 0 prime. Alternatively, and due to low yield of selected recombinant NACHOs, ferrets were boosted on day 28 with 15 µg of homologous NACHO (A/Perth/16/2009nib-64, A/Tasmania/1018/2015, A/Utah/11/2011, and A/West Virginia/17/2012) or were inoculated with the same H1N2 reassortant virus used in the prime (A/Estonia/91621/2015 and A/Stockholm/15/2014). Sera from bleeds 2 weeks after the prime (day 14) and booster vaccination (day 42) were tested for immunogenicity and seroconversion in serological assays.

The mouse immunisation experiments were conducted according to the Belgian legislation (Belgian Law 14/08/1986 and Belgium Royal Decree 06/04/2010) and European legislation on protection of animals used for scientific purposes (EU directives 2010/63/EU and 86/609/EEC). Experimental protocols were all approved by the Ethics Committee of the Vlaams Instituut voor Biotechnologie (VIB), Ghent University, Faculty of Science (EC2019-035). Female BALB/c mice, aged 6–7 weeks, were purchased from Charles River. The mice were housed in a specified pathogen-free animal house with food and water ad libitum. Animals were immunised intramuscularly in the right quadriceps. Immunisation was performed with 50 µl containing 1 µg of recombinant protein (tetrabrachion stabilised NAs, referred to as NACHOs). All, except A/West Virginia/17/2012 NACHOs, were used as immunogens in mice. All immunisations were adjuvanted with a 1:1 volume of AF03 (25 µl of antigen in PBS + 25 µl of AF03 per dose) (*Caillet et al., 2010*).

Two or three weeks after the boost immunisation, mice were humanely sacrificed with an anaesthetic overdose of Nembutal (6 mg per mouse) and bled retro-orbitally. The obtained blood samples were incubated for 30 min at 37°C to allow clotting, which was followed by centrifugation at 10,500×*g* for 5 min. The supernatant (the serum) was recovered and submitted to a second centrifugation at 10,500×*g* for 5 min. Cleared sera were stored at –20°C before use in serological assays.

## ELISA

Anti-tetNA IgG levels in mouse and ferret sera were determined by capture ELISA using tetNAs in Pierce nickel-coated plates (cat. # 15442). Recombinant proteins were diluted in DPBS (Life Technologies cat. # 14040-182) (tetNA was diluted to 0.5 µg/ml). Then, 50 µl of the coating antigen solutions was added to each well and the plate was incubated at room temperature for 1 hr on a shaking platform. The wells of the plates were then washed three times with PBS-T (Sigma cat. # P3563-10PAK) and blocked for 1 hr with 1% BSA (for the mouse sera) or 10% milk (for the ferret sera) in DPBS. After blocking, the wells of the plates were washed once with PBS-T and incubated with a threefold serial dilution, starting at a 1/100 dilution, of serum in DPBS, 0.5% BSA, 0.05% Tween-20 for 2 hr at room temperature on a shaking platform. The wells of the plates were then washed five times with PBS-T and incubated with a 1:5000 dilution of anti-mouse IgG-HRP (GE Healthcare cat. # NA931-1ml), or with a 1:100,000 dilution of anti-ferret IgG-HRP (Abcam cat. # 112770) in DPBS, 0.5% BSA, 0.05%

Tween-20. The 3,3′,5,5′-tetramethylbenzidine (TMB) substrate (BD cat. # 555214) was added after three washes with PBS-T and the reaction was stopped after 5 min by addition of 50 µl of 1 M $H_2SO_4$. The optical density (OD) in each well was determined at 450 nm and, as a reference, 655 nm using an iMark Microplate Absorbance Reader (Bio-Rad). The end point titer was determined for each serum sample by scoring the dilution that resulted in an OD that was equal to or two times higher than the background OD obtained from pre-immune control sera dilution series.

## ELLA to determine NAI titers

Fetuin (Sigma cat. # F3385) was diluted into coating buffer (KPL cat. # 50-84-01) to a concentration of 25 µg/ml and 50 µl was added to the wells of Nunc MaxiSorp plates (Thermo Fisher cat. # 44-2404-21), which were incubated overnight at 4°C. The coated plates were then washed three times with PBS-T (Sigma cat. # P3563-10PAK) and incubated overnight with 60 µl of a dilution of H6N2 and 60 µl of twofold serial dilution of serum, starting at a 1/20 dilution, in sample buffer (1× MES VWR cat. # AAJ61979-AP: 20 mM $CaCl_2$, 1% BSA, 0.5% Tween-20). These dilutions of the H6Nx viruses correspond to the 70% maximum activity of NA from the respective viruses as determined in the ELLA. Fetuin-coated plates were then washed three times with PBS-T and incubated for 1 hr with a solution of PNA-HRP (cat. # L6135-1MG, Sigma) at 5 µg/ml in conjugate diluent (MES pH 6.5, 20 mM $CaCl_2$, 1% BSA). The plates were washed three times with PBS-T, TMB substrate was added, and then the plates were incubated for 5 min before the reaction was stopped by the addition of 50 µl of 1 M $H_2SO_4$. The optical density was measured at 450 nm and, as a reference, 655 nm in an iMark Microplate Absorbance Reader (Bio-Rad). Half maximum inhibitory concentrations ($IC_{50}$) values were determined by non-linear regression analysis (GraphPad Prism software).

## NA enzyme kinetics assay

The enzymatic activity of NACHOs and H6N2s was determined with the fluorogenic small substrate MUNANA as previously described (*Marathe et al., 2013*). First H6N2 dilutions were tested to ensure that MUNANA conversion into 4-methylumbelliferone (4-MU) generated fluorescence over time at a linear ratio. Ten microliters of 1 mM of MUNANA (Sigma cat. # M8639) in 200 mM sodium acetate buffer (pH 6.5) containing 2 mM $CaCl_2$ and 1% butanol were incubated with 40 µl of PBS solution containing viral dilution. Conversion to 4-MU was monitored every 2 min for 1 hr using a BMG Fluostar OPTIMA reader (excitation at 365 nm and emission determined at 450 nm). Second, using the selected dilution of virus, the assay was repeated with dilutions of MUNANA ranging from 2.5 to 100 µM. A standard curve with 800 to 25 pmoles/well of 4-MU (Sigma cat. # M1508) was used to extrapolate the molar conversion of MUNANA. $K_M$ was calculated using GraphPad Prism version 8.4.3.

## Antigenic cartography

Antigenic cartography was performed as described by *Smith et al., 2004*, using the R CRAN package (https://CRAN.R-project.org/package=Racmacs). The antigenic map was optimised 5000 times and the best optimisation was kept (stress value = 1366.47). Uncertainty was measured by bootstrapping 1000 repeats with 100 optimisations per repeat.

## Association study

For the association analysis, under the assumption of homogeneous genetic relatedness among the selected H3N2 virus strains, a simple linear regression model for mutation-trait association analysis was fitted to the data for each H6N2 separately. The significance values were calculated by comparing $IC_{50}$ values of sera that were classified into two groups according to the presence or absence of variation in the amino acid of reference Tex12 sequence.

## Correlations with distances from the NA catalytic pocket

Tau values were determined by correlating distances with significance values obtained from the association analysis or the importance scores obtained from machine learning. A similar method was described by Meyer and Wilke to define geometrical constraints of HA evolution (*Meyer and Wilke, 2015*). Briefly, the relative distances of each amino acids to the 98 variable residues present in our N2 pnael were calculated based on the structure of A/Perth/2009 (H3N2) NA (PDB: 6BR6) (*Hadházi et al., 2018*). The distances were then correlated with the significance values that were obtained in

the association analysis, or the importance scores obtained in the machine learning. Correlation coefficients were visualised on the structure of N2 NA derived from pandemic A/RI/5+/1957 H2N2 virus (PDB: 4H53) (*Vavricka et al., 2013*) using Chimera 1.14.

## RF algorithm

The RF algorithm was adapted from *Yao et al., 2017*. Briefly, the non-conserved positions in the multiple sequence alignment of N2 head domain amino acid sequences of all tested N2s were selected. The feature matrix ($X$) was created by assessing matches/mismatches on each non-conserved position for each pair of N2 head domain amino acid sequences as:

$$X = \begin{array}{c} \\ V_1V_2 \\ \vdots \\ V_1V_n \\ \vdots \\ V_{n-1}V_n \end{array} \begin{array}{cccc} X_1 & X_2 & \ldots & X_m \\ 0 & 1 & \ldots & 1 \\ \vdots & \vdots & \ldots & \vdots \\ 1 & 0 & \ldots & 1 \\ \vdots & \vdots & \ldots & \vdots \\ 1 & 0 & \ldots & 1 \end{array}$$

where $X_i$ represents the non-conserved positions in the multiple sequence alignment, $m$ is the number of non-conserved positions (102 in total), $V_i$ represents the N2 head domain sequence for each tested NA $n$ is the number of tested sera (43 in total), 0 represents a match, 1 represents a mismatch in the alignment.

The response vector ($Y$) is composed of the antigenic distances for each pair of tested sera. These antigenic distances were determined based on NAI, similar to the method used to determine HI-based immunogenic distances that was described by Cai et al. In summary, the distances between sera were calculated as $\frac{1}{n}\sum_{t=1}^{n} d[t,i] - d[i,t]$, which represent average distances between two serum samples among all tested H6N2s (*Cai et al., 2012*). We applied the RF function from the randomForest package in R to construct the model. We set the bootstrapping (tree) number to 1500 and the number of features to 25 for each tree. Accuracy was determined as the RMSE, calculated using the RMSE function from the yardstick package in R.

Additionally, predicted antigenic distances of the out-of-bag samples were compared to the observed distances by fitting a linear model and calculating the $R^2$ value. To calculate the importance of each feature ($X_i$), the following metrics were calculated using the measure importance function of the randomForestExplainer package in R:

- Mean minimal depth: mean of the depths of the node in each tree that splits on this feature and is the closest to the root of the tree (minimal depth). The lower, the more important the feature since a low depth means that many observations are divided into groups based on this feature.
- mse decrease: mean decrease of accuracy in predictions on out-of-bag samples when the feature is excluded from the model. The higher, the more important the feature.

The data and code used for the generation of the RF model is available at https://github.com/SaelensLAB/RF, copy archived at *SaelensLAB, 2024*.

The data and code used for the generation of the RF model is available at https://github.com/SaelensLAB/RF, copy archived at *SaelensLAB, 2024*.

## Acknowledgements

We gratefully acknowledge all data contributors, i.e., the authors and their Originating laboratories responsible for obtaining the specimens, and their Submitting laboratories for generating the genetic sequence and metadata and sharing via the GISAID Initiative, on which this research is based. We are grateful to the animal caretakers of the animal house at the VIB-UGent Center for Inflammation Research. This work was funded by Sanofi Vaccines.

# Additional information

## Competing interests

Guadalupe Cortes-Garcia, Tod Strugnell, Raul Gomila, John Hamberger, John Catalan, Irina V Ustyugova, Timothy Farrell, Svetlana Stegalkina, Satyajit Ray, Lauren LaRue, Thorsten U Vogel: Sanofi employee and may hold stock options in the company. The other authors declare that no competing interests exist.

## Funding

| Funder | Grant reference number | Author |
|---|---|---|
| Sanofi | | Guadalupe Cortes-Garcia<br>Tod Strugnell<br>Raul Gomila<br>John Hamberger<br>John Catalan<br>Irina V Ustyugova<br>Timothy Farrell<br>Svetlana Stegalkina<br>Satyajit Ray<br>Lauren LaRue<br>Xavier Saelens<br>Thorsten U Vogel |
| Vlaams Instituut voor Biotechnologie | | João Paulo Portela Catani<br>Anouk Smet<br>Tine Ysenbaert<br>Marnik Vuylsteke<br>Guy Bottu<br>Janick Mathys<br>Alexander Botzki |

The funders had no role in study design, data collection and interpretation, or the decision to submit the work for publication.

## Author contributions

João Paulo Portela Catani, Conceptualization, Data curation, Formal analysis, Funding acquisition, Investigation, Methodology, Writing – original draft, Writing – review and editing; Anouk Smet, Tine Ysenbaert, Formal analysis, Investigation, Methodology, Writing – review and editing; Marnik Vuylsteke, Guy Bottu, Janick Mathys, Alexander Botzki, Tod Strugnell, Formal analysis; Guadalupe Cortes-Garcia, Satyajit Ray, Investigation, Writing – review and editing; Raul Gomila, Irina V Ustyugova, Writing – review and editing; John Hamberger, John Catalan, Timothy Farrell, Lauren LaRue, Project administration; Svetlana Stegalkina, Investigation; Xavier Saelens, Conceptualization, Formal analysis, Supervision, Funding acquisition, Methodology, Writing – original draft, Project administration, Writing – review and editing; Thorsten U Vogel, Conceptualization, Resources, Formal analysis, Supervision, Funding acquisition, Project administration, Writing – review and editing

## Author ORCIDs

João Paulo Portela Catani https://orcid.org/0000-0002-4866-3933
Janick Mathys http://orcid.org/0009-0007-1722-2370
Alexander Botzki http://orcid.org/0000-0001-6691-4233
Irina V Ustyugova https://orcid.org/0000-0001-6856-4433
Xavier Saelens http://orcid.org/0000-0002-3861-6965
Thorsten U Vogel https://orcid.org/0000-0002-0371-8048

## Ethics

The ferret immunization experiment was conducted at Bioqual, Inc, MD (USA). The experiments were approved by the Institutional Animal Care and Use Committee of the Bioqual (reference number 17-021.6). Outbred 18-22 weeks old native male and female Fitch ferrets were obtained from Triple F, PA (USA) and housed under specified pathogen-free conditions with food and water ad libitum. The mouse immunization experiments were conducted according to the Belgian legislation (Belgian Law 14/08/1986 and Belgium Royal Decree 06/04/2010) and European legislation on protection of animals

used for scientific purposes (EU directives 2010/63/EU and 86/609/EEC). Experimental protocols were all approved by the Ethics Committee of the Vlaams Instituut voor Biotechnologie (VIB), Ghent University, Faculty of Science (EC2019-035). Female BALB/c mice, aged 6-7 weeks, were purchased from Charles River. The mice were housed in a specified pathogen-free animal house with food and water ad libitum.

Reviewer #1 (Public review): https://doi.org/10.7554/eLife.90782.4.sa1
Reviewer #2 (Public review): https://doi.org/10.7554/eLife.90782.4.sa2
Reviewer #3 (Public review): https://doi.org/10.7554/eLife.90782.4.sa3
Author response https://doi.org/10.7554/eLife.90782.4.sa4

## Additional files

### Supplementary files
• Supplementary file 1. Specific activity of the recombinant NACHOs.

• Supplementary file 2. H6N2 reassortant viruses used in the enzyme-linked lectin assay (ELLA) assay.

• Supplementary file 3. Neuraminidase inhibition (NAI) titers against H6N2 viruses of ferret sera obtained after the boost.

• MDAR checklist

### Data availability
All data and source code are available within this manuscript.

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
