## [Editor Report · eLife assessment]

This study presents **valuable** data on the antigenic properties of neuraminidase proteins of human A/H3N2 influenza viruses sampled between 2009 and 2017. The antigenic properties are found to be generally concordant with genetic groups. Compared to a previous version, additional analyses have strengthened the work, with **solid** evidence supporting the claims of the authors.

---

## [Referee Report · Reviewer #1 (Public review)]

Summary

The authors investigated the antigenic diversity of recent (2009-2017) A/H3N2 influenza neuraminidases (NAs), the second major antigenic protein after haemagglutinin. They used 27 viruses and 43 ferret sera and performed NA inhibition. This work was supported by a subset of mouse sera. Clustering analysis determined 4 antigenic clusters, mostly in concordance with the genetic groupings. Association analysis was used to estimate important amino acid positions, which were shown to be more likely close to the catalytic site. Antigenic distances were calculated and a random forest model used to determine potential important sites.

This revision has addressed many of my concerns of inconsistencies in the methods, results and presentation. There are still some remaining weaknesses in the computational work.

Strengths

(1) The data cover recent NA evolution and a substantial number (43) of ferret (and mouse) sera were generated and titrated against 27 viruses. This is laborious experimental work and is the largest publicly available neuraminidase inhibition dataset that I am aware of. As such, it will prove a useful resource for the influenza community.

(2) A variety of computational methods were used to analyse the data, which give a rounded picture of the antigenic and genetic relationships and link between sequence, structure and phenotype.

(3) Issues raised in the previous review have been thoroughly addressed.

Weaknesses:

Some concerns regarding the robustness of the machine learning model and potential overfitting remain.

---

## [Referee Report · Reviewer #2 (Public review)]

Summary:

The authors characterized the antigenicity of N2 protein of 43 selected A(H3N2) influenza A viruses isolated from 2009-2017 using ferret and mice immune sera. Four antigenic groups were identified, which the authors claimed to be correlated with their respective phylogenic/ genetic groups. Among 102 amino acids differed by the 44 selected N2 proteins, the authors identified residues that differentiate the antigenicity of the four groups and constructed a machine-learning model that provides antigenic distance estimation. Three recent A(H3N2) vaccine strains were tested in the model but there was no experimental data to confirm the model prediction results.

Strengths:

This study used N2 protein of 44 selected A(H3N2) influenza A viruses isolated from 2009-2017 and generated corresponding panels of ferret and mouse sera to react with the selected strains. The amount of experimental data for N2 antigenicity characterization is large enough for model building.

Weaknesses:

One weakness is the use of double-immune ferret sera (post-infection plus immunization with recombinant NA protein) or mouse sera (immunized twice with recombinant NA protein) to characterize the antigenicity of the selected A(H3N2) viruses. Conventionally, NA antigenicity is characterized using ferret sera after a single infection. Repeated influenza exposure in ferrets has been shown to enhance antibody binding affinity and may affect the cross-reactivity to heterologous strains (PMID: 29672713). The increased cross-reactivity is supported by the NAI titers shown in Table S3, as many of the double immune ferret sera showed the highest reactivity not against its own homologous virus but to heterologous strains. In response to the reviewer's comment, the authors agreed the use of double-immune ferret sera may be a limitation of the study.

Another weakness is that the authors used the newly constructed a model to predict antigenic distance of three recent A(H3N2) viruses but there is no experimental data to validate their prediction (eg. if these viruses are indeed antigenically deviating from group 2 strains as concluded by the authors). Leaving out data from some strains for testing is a useful check, but due to phylogenetic correlations in the data the generalizability of the machine learning is not guaranteed.

---

## [Referee Report · Reviewer #3 (Public review)]

Summary:

This paper by Portela Catani et al examines the antigenic relationships (measured using monotypic ferret and mouse sera) across a panel of N2 genes from the past 14 years, along with the underlying sequence differences and phylogenetic relationships. This is a highly significant topic given the recent increased appreciation of the importance of NA as a vaccine target, and the relative lack of information about NA antigenic evolution compared with what is known about HA. Thus, these data will be of interest to those studying the antigenic evolution of influenza viruses. The methods used are generally quite sound, though there are a few addressable concerns that limit the confidence with which conclusions can be drawn from the data/analyses.

Strengths:

-The significance of the work, and the (general) soundness of the methods.

-Explicit comparison of results obtained with mouse and ferret sera

Weaknesses:

- Machine learning analyses neither experimentally validated nor shown to be better than simple, phylogenetic-based inference.

---

## [Author Response]

The following is the authors’ response to the previous reviews.

**eLife assessment**
This study presents valuable data on the antigenic properties of neuraminidase proteins of human A/H3N2 influenza viruses sampled between 2009 and 2017. The antigenic properties are found to be generally concordant with genetic groups. Additional analysis have strengthened the revised manuscript, and the evidence supporting the claims is solid.
**Public Reviews:**

**Reviewer #1 (Public Review):**
SummaryThe authors investigated the antigenic diversity of recent (2009-2017) A/H3N2 influenza neuraminidases (NAs), the second major antigenic protein after haemagglutinin. They used 27 viruses and 43 ferret sera and performed NA inhibition. This work was supported by a subset of mouse sera. Clustering analysis determined 4 antigenic clusters, mostly in concordance with the genetic groupings. Association analysis was used to estimate important amino acid positions, which were shown to be more likely close to the catalytic site. Antigenic distances were calculated and a random forest model used to determine potential important sites.This revision has addressed many of my concerns of inconsistencies in the methods, results and presentation. There are still some remaining weaknesses in the computational work.Strengths(1) The data cover recent NA evolution and a substantial number (43) of ferret (and mouse) sera were generated and titrated against 27 viruses. This is laborious experimental work and is the largest publicly available neuraminidase inhibition dataset that I am aware of. As such, it will prove a useful resource for the influenza community.(2) A variety of computational methods were used to analyse the data, which give a rounded picture of the antigenic and genetic relationships and link between sequence, structure and phenotype.(3) Issues raised in the previous review have been thoroughly addressed.Weaknesses(1). Some inconsistencies and missing data in experimental methodsTwo ferret sera were boosted with H1N2, while recombinant NA protein for the others. This, and the underlying reason, are clearly explained in the manuscript. The authors note that boosting with live virus did not increase titres. Additionally, one homologous serum (A/Kansas/14/2017) was not generated, although this would not necessarily have impacted the results.

We agree with the reviewer and this point was addressed in the previous rebuttal.

(2) Inconsistency in experimental resultsClustering of the NA inhibition results identifies three viruses which do not cluster with their phylogenetic group. Again this is clearly pointed out in the paper and is consistent with the two replicate ferret sera. Additionally, A/Kansas/14/2017 is in a different cluster based on the antigenic cartography vs the clustering of the titres

We agree with the reviewer and this point was addressed in the previous rebuttal.

(3) Antigenic cartography plot would benefit from documentation of the parameters and supporting analyses

a. The number of optimisations used

We used 500 optimizations. This information is now included in the Methods section.

b. The final stress and the difference between the stress of the lowest few (e.g. 5) optimisations, or alternatively a graph of the stress of all the optimisations. Information on the stress per titre and per point, and whether any of these were outliers

The stress was obtained from 1, 5, 500, or even 5000 optimizations (resulting in stress values of respectively, 1366.47, 1366.47, 2908.60, and 3031.41). Besides limited variation or non-conversion of the stress values after optimization, the obtained maps were consistent in multiple runs. The map was obtained keeping the best optimization (stress value 1366.47, selected using the keepBestOptimization() function).

**Author response image 1. sa4fig1:** The stress per point is presented in the heat map below.

The heat map indicates stress per serum (x-axis) and strain (y-axis) in blue to red scale.

c. A measure of uncertainty in position (e.g. from bootstrapping)

Bootstrap was performed using 1000 repeats and 100 optimizations per repeat. The uncertainty is represented in the blob plot below.

**Author response image 2. sa4fig2:** 

(4) Random forestThe full dataset was used for the random forest model, including tuning the hyperparameters. It is more robust to have a training and test set to be able to evaluate overfitting (there are 25 features to classify 43 sera).

Explicit cross validation is not necessary for random forests as the out of bag process with multiple trees implicitly covers cross validation. In the random forest function in R this is done by setting the mtry argument (number of variables randomly sampled as candidates at each split). R samples variables with replacement (the same variable can be sampled multiple times) of the candidates from the training set. RF will then automatically take the data that is not selected as candidates as test set. Overfit may happen when all data is used for training but the RF method implicitly does use a test set and does not use all data for training.

Code:

rf <- randomForest(X,y=Y,ntree=1500,mtry=25,keep.forest=TRUE,importance=TRUE)

**Reviewer #2 (Public Review):**

**Summary:**
The authors characterized the antigenicity of N2 protein of 43 selected A(H3N2) influenza A viruses isolated from 2009-2017 using ferret and mice immune sera. Four antigenic groups were identified, which the authors claimed to be correlated with their respective phylogenic/ genetic groups. Among 102 amino acids differed by the 44 selected N2 proteins, the authors identified residues that differentiate the antigenicity of the four groups and constructed a machine-learning model that provides antigenic distance estimation. Three recent A(H3N2) vaccine strains were tested in the model but there was no experimental data to confirm the model prediction results.Strengths:This study used N2 protein of 44 selected A(H3N2) influenza A viruses isolated from 2009-2017 and generated corresponding panels of ferret and mouse sera to react with the selected strains. The amount of experimental data for N2 antigenicity characterization is large enough for model building.Weaknesses:The main weakness is that the strategy of selecting 43 A(H3N2) viruses from 2009-2017 was not explained. It is not clear if they represent the overall genetic diversity of human A(H3N2) viruses circulating during this time. In response to the reviewer's comment, the authors have provided a N2 phylogenetic tree using180 randomly selected N2 sequences from human A(H3N2) viruses from 2009-2017. While the 43 strains seems to scatter across the N2 tree, the four antigenic groups described by the author did not correlated with their respective phylogenic/ genetic groups as shown in Fig. 2. The authors should show the N2 phylogenic tree together with Fig. 2 and discuss the discrepancy observed.

The discrepancies between the provided N2 phylogenetic tree using 180 selected N2 sequences was primarily due to visualization. In the tree presented in Figure 2 the phylogeny was ordered according to branch length in a decreasing way. Further, the tree represented in the rebuttal was built with PhyML 3.0 using JTT substitution model, while the tree in figure 2 was build in CLC Workbench 21.0.5 using Bishop-Friday substitution model. The tree below was built using the same methodology as Figure 2, including branch size ordering. No discrepancies are observed.

Phylogenetic tree representing relatedness of N2 head domain. N2 NA sequences were ordered according to the branch length and phylogenetic clusters are colored as follows: G1: orange, G2: green, G3: blue, and G4: purple. NA sequences that were retained in the breadth panel are named according to the corresponding H3N2 influenza viruses. The other NA sequences are coded.

**Author response image 3. sa4fig3:** 

The second weakness is the use of double-immune ferret sera (post-infection plus immunization with recombinant NA protein) or mouse sera (immunized twice with recombinant NA protein) to characterize the antigenicity of the selected A(H3N2) viruses. Conventionally, NA antigenicity is characterized using ferret sera after a single infection. Repeated influenza exposure in ferrets has been shown to enhance antibody binding affinity and may affect the cross-reactivity to heterologous strains (PMID: 29672713). The increased cross-reactivity is supported by the NAI titers shown in Table S3, as many of the double immune ferret sera showed the highest reactivity not against its own homologous virus but to heterologous strains. In response to the reviewer's comment, the authors agreed the use of double-immune ferret sera may be a limitation of the study. It would be helpful if the authors can discuss the potential effect on the use of double-immune ferret sera in antigenicity characterization in the manuscript.

Our study was designed to understand the breadth of the anti-NA response after the incorporation of NA as a vaccine antigens. Our data does not allow to conclude whether increased breadth of protection is merely due to increased antibody titers or whether an NA boost immunization was able to induce antibody responses against epitopes that were not previously recognized by primary response to infection. However, we now mention this possibility in the discussion and cite Kosikova et al. CID 2018, in this context.

Another weakness is that the authors used the newly constructed a model to predict antigenic distance of three recent A(H3N2) viruses but there is no experimental data to validate their prediction (eg. if these viruses are indeed antigenically deviating from group 2 strains as concluded by the authors). In response to the comment, the authors have taken two strains out of the dataset and use them for validation. The results is shown as Fig. R7. However, it may be useful to include this in the main manuscript to support the validity of the model.

The removal of 2 strains was performed to illustrate the predictive performance of the RF modeling. However, Random Forest does not require cross-validation. The reason is that RF modeling already uses an out-of-bag evaluation which, in short, consists of using only a fraction of the data for the creation of the decision trees (2/3 of the data), obviating the need for a set aside the test set:

“…In each bootstrap training set, about one-third of the instances are left out. Therefore, the out-of-bag estimates are based on combining only about one- third as many classifiers as in the ongoing main combination. Since the error rate decreases as the number of combinations increases, the out-of-bag estimates will tend to overestimate the current error rate. To get unbiased out-of-bag estimates, it is necessary to run past the point where the test set error converges. But unlike cross-validation, where bias is present but its extent unknown, the out-of-bag estimates are unbiased…” from https://www.stat.berkeley.edu/%7Ebreiman/randomforest2001.pdf

**Reviewer #3 (Public Review):**
Summary:This paper by Portela Catani et al examines the antigenic relationships (measured using monotypic ferret and mouse sera) across a panel of N2 genes from the past 14 years, along with the underlying sequence differences and phylogenetic relationships. This is a highly significant topic given the recent increased appreciation of the importance of NA as a vaccine target, and the relative lack of information about NA antigenic evolution compared with what is known about HA. Thus, these data will be of interest to those studying the antigenic evolution of influenza viruses. The methods used are generally quite sound, though there are a few addressable concerns that limit the confidence with which conclusions can be drawn from the data/analyses.Strengths:The significance of the work, and the (general) soundness of the methods.-Explicit comparison of results obtained with mouse and ferret seraWeaknesses:Approach for assessing influence of individual polymorphisms on antigenicity does not account for potential effects of epistasis (this point is acknowledged by the authors).

We agree with the reviewer and this point was addressed in the previous rebuttal.

Machine learning analyses neither experimentally validated nor shown to be better than simple, phylogenetic-based inference.

We respectfully disagree with the reviewer. This point was addressed in the previous rebuttal as follows.

This is a valid remark and indeed we have found a clear correlation between NAI cross reactivity and phylogenetic relatedness. However, besides achieving good prediction of the experimental data (as shown in Figure 5 and in FigureR7), machine Learning analysis has the potential to rank or indicate major antigenic divergences based on available sequences before it has consolidated as new clade. ML can also support the selection and design of broader reactive antigens. “

**Recommendations for the authors:**

**Reviewer #2 (Recommendations For The Authors):**
(1) Discuss the discrepancy between Fig. 2 and the newly constructed N2 phylogenetic tree with 180 randomly selected N2 sequences of A(H3N2) viruses from 2009-2017. Specifically please explain the antigenic vs. phylogenetic relationship observed in Fig. 2 was not observed in the large N2 phylogenetic tree.

Discrepancies were due to different method and visualization. A new tree was provided.

(2) Include a sentence to discuss the potential effect on the use of double-immune ferret sera in antigenic characterization.

We prefer not to speculate on this.

(3) Include the results of the exercise run (with the use of Swe17 and HK17) in the manuscript as a way to validate the model.

The exercise was performed to illustrate predictive potential of the RF modeling to the reviewer. However, cross-validation is not a usual requirement for random forest, since it uses out-of-bag calculations. We prefer to not include the exercise runs within the main manuscript.